# The impact of tax accounting and planning on earnings management: Evidence from panel ARDL approach

Meral Gündüz*

Department of Accounting and Finance Management, Uşak University, Uşak, Türkiye

* meral.gunduz@usak.edu.tr

## Abstract

Some companies may mislead stakeholders by using the flexibility in accounting standards when determining the amount of profit to be disclosed, a practice referred to as earnings management. Deferred taxes are one of the flexibilities that enable this practice. This study contributes to the growing literature on earnings management in private companies by focusing on deferred taxes and tax planning. The purpose of this study is to investigate the relationship between deferred tax assets, deferred tax expense and tax planning of companies with earnings management. Data from companies listed in the Borsa Istanbul BIST 30 index in Türkiye for 2013–2022 are analyzed using panel data methods. The study results show that deferred tax expenses and deferred tax assets negatively impact earnings management. The analysis including tax planning reveals that deferred tax expenses, deferred tax assets and tax planning also have a negative impact on earnings management. However, tax planning reduces the impact of deferred tax expense on earnings management.

## 1. Introduction

In a competitive environment, firms endeavor to secure an advantage over their rivals primarily through sales revenue generation and the enhancement of product quality. To this end, managers seek to establish rigorous and strategically oriented decision-making processes [1]. Ultimately, the pursuit of superior profitability emerges as the most critical objective in sustaining their competitive edge [2]. For this reason, some company managements sometimes perform earnings management during the preparation of financial statements for a specific purpose [3,4]. Earnings management is an endeavour by managers to influence the information in financial statements in order to mislead stakeholders who want to be informed about the company's performance and financial position [5]. Hence, earnings management involves the deviations made by management in accounting practices to reassure stakeholders

**Data availability statement:** All relevant data are within the paper and its Supporting Information files.

**Funding:** The author(s) received no specific funding for this work.

**Competing interests:** The author has declared that no competing interests exist.

while ensuring the company achieves its financial reporting objectives in accordance with normal business activities [2,6,7]. Earnings management refers to tendency of managers to influence earnings for the relevant year by manipulating real operations [8]. Managers use the flexibility or gaps in accounting standards and principles to implement earnings management practices. With these practices, managers use the flexibilities in accounting principles and procedures to determine the profit in line with financial information users' expectations [9].

Managers manage accounting profit by employing earnings management within legal limits to achieve specific objectives. They make the profit look different through various accounting policies and the classification of expenses and revenues. Three main earnings management techniques are recognised in the accounting literature: accrual management, activity management and classification method [10]. Earnings management by accrual management is the most researched and popular approach. In this method, profit for the period is managed using the accounting policies they determine with the flexibility opportunities provided by accounting standards [11]. Accounting standards provide managers with considerably more flexible in determining accounting principles and assumptions than tax legislation allows [12].

Legal frameworks, particularly in multinational corporate groups, exert a significant influence on the concrete choice of legal structure. Consequently, selecting a legally optimized structure can serve as a tool for tax planning and internal financing [13]. Deferred taxes are one of the flexibilities used in accrual management in earnings management practices. Management can manipulate earnings when determining the value of deferred tax expenses (DTE). This manipulation is performed by changing the constituents of deferred tax assets (DTA) or deferred tax liabilities (DTL) [14]. This type of transaction is often expressed as tax planning and is a widespread strategy companies use to pay less tax. DTAs are used to reduce income tax. However, DTE reverse this situation and increase tax expenses that is, management can manipulate earnings using DTE. In particular, DTE allow the acceleration of expenses, allowing companies to delay revenues and save tax. Furthermore, DTE allow management to increase income without paying extra taxes in the current period. This approach allows the company to increase profits by deferring its tax liability [15].

In Türkiye, the Turkish Accounting Standards (TMS) 12 Income Taxes Standard was published in 2006 in line with the International Accounting Standards (IAS) prepared by the International Accounting Standards Board (IASB). This standard regulates the accounting for income taxes (taxes calculated on corporate income) [16]. The standard also deals with the accounting of DTA or unused tax advantages arising from prior year losses that have not yet been used, the presentation of income taxes in financial statements, and the disclosure of information regarding income taxes. In TMS 12 Income Taxes, DTA and DTL are recognized. DTA represent the tax amount of an entity will recover in the time to come. Hence, DTA are the amounts arising from enterprises' deductible temporary differences or tax advantages, which cannot be deducted from the tax burden in the current period but can be deducted in the future period. DTL, on the other hand, represent the income taxes that will be payable in future periods based on taxable temporary differences [17,18].

In Türkiye, companies prepare financial statements based on Turkish Financial Reporting Standards, Turkish Accounting Standards, and tax laws. For this reason, the commercial profit amounts calculated by the companies and the financial profit amounts may differ. These differences can be permanent or temporary. In the case of temporary differences between commercial and financial profit, the concept of deferred tax comes into play. Temporary differences are classified into deductible temporary differences and taxable temporary differences. Deductible temporary differences produce a deferred tax asset, and taxable temporary differences produce a deferred tax liability [19].

IAS 12 income taxes partly limit the recognition of DTA to the generation of taxable profits from which the tax benefits contained in those DTA can be utilized. Therefore, the recognition of a DTA give a strong signal to external users about the probability that the companies will produce taxable profits in the future. Given the strong connection between accounting profit and taxable profit, the recognition of a deferred tax asset indicates that future accounting profits are expected to be realized [20]. Therefore, DTA are one of the practices that provide opportunities for earnings management. DTL arise when taxable income is less than pre-tax accounting income. The reason for DTA is that taxable income exceeds pre-tax accounting income [3].

Although research on earnings management has increased in recent years [3], research on the relationship between deferred taxes and tax planning (TP) and earnings management remains relatively unexplored [21]. Managers can engage in earnings management by using deferred taxes and tax planning to affect book income without affecting taxable income [11]. Thus, the objective of this study is to examine the relationship between tax accounting and planning and earnings management in companies listed on Borsa Istanbul (BIST) in Türkiye.

This study contributes to literature in three main ways. First, research that simultaneously examines the effects of DTE, DTA and TP on earnings management is quite limited. Literature generally investigates these variables separately [2,14,22]. In this study, however, the effects of DTE and DTA on earnings management are analyzed together, and subsequently, the analysis is repeated by including the tax planning factor in the model. In this context, the findings provide empirical evidence that these three variables are strong predictors of tax-driven earnings management. Second, the study is unique in that it simultaneously analyzes variables that may have opposing effects on earnings management. TP is a mechanism that firms use to pay less tax, benefit from tax advantages, and reduce future tax liabilities, thus exerting a profit-reducing effect, whereas DTE allow management to increase profits in the same period without additional tax payments and can therefore be used for earnings manipulation [23,24]. DTA, on the other hand, reduce the current period tax burden and indirectly enable profit management. In this regard, the study provides a comprehensive perspective on how managers may use or prefer these tools together, contributing to a better understanding of tax management dynamics within firms. Third, most existing studies have been conducted in the United States, China, Indonesia, and European countries [11, 25,26]. In Türkiye, there is no empirical study, to the best of the researcher's knowledge, that simultaneously examines the relationship between DTE, DTA, TP and earnings management. Therefore, this research fills the gap in the Turkish context and provides comparative contributions to the international literature.

This research is structured into five parts. In the introductory part, the concepts of earnings management, DTA, DTL, DTE, and TP are explained, and information about the TAS 12 Income Taxes Standard in Türkiye is provided. The second part provides a literature review of relevant studies. The third part provides information on data and methodology. The fourth part presents empirical analysis and findings. The fifth part discusses the findings and provides policy recommendations.

## 2. Literature review

The earnings management literature traces its origins to pioneering studies such as Healy [27], DeAngelo [28], and McNichols and Wilson [29], and underwent a significant methodological transformation with Jones [30]. The expected accruals model developed by Jones [30], subsequently extended through variations by DeFond and Jiambalvo [31], Dechow, Sloan, and Sweeney [32], and Kothari, Leone, and Wasley [33], has become one of the most widely adopted

measurement tools in contemporary research. This foundational literature has provided the basis for examining earnings management, particularly through accrual-based models.

Over time, alongside accrual models, variables related to taxation have emerged as important indicators for explaining earnings management. Phillips et al. [14] is one of the pioneering studies showing that DTE offers additional explanatory power for detecting earnings management aimed at mitigating earnings declines and preventing losses beyond total accruals. Christensen et al. [34] examined the use of deferred tax valuation allowances in "big bath" practices yet found no strong evidence that these allowances are reversed in subsequent periods to increase earnings. These findings have initiated academic debate regarding the relationship between deferred tax items and earnings management.

Empirical studies, however, show highly inconsistent results. On one hand, there are studies supporting a significant relationship between DTE and earnings management. Kisno and Istianingsih [22], demonstrated that deferred tax expense, together with leverage, positively affects earnings management, while Handayani et al. [24] found a positive and significant impact of deferred taxes. Nurfadila and Muslim [35] identified a positive correlation between DTE associated with tax planning and earnings management, and Machdar and Nurdiniah [2] showed that DTA and DTE influence accrual-based earnings management. Similarly, Permatasari and Trisnawati [6] argue that DTE positively affects earnings management.

Conversely, some studies do not support this relationship or reveal different aspects. Yacob et al. [36] found that deferred taxes were not used as a tool for earnings management in Malaysia. Bunaca [37] reported that DTE affects earnings management but not profitability, whereas tax planning affects profitability but not earnings management. Purnamasari [38] and Fajarwati et al. [3] observed that DTA and DTE have no significant effect on earnings management. Rachmany and Tajudin [39] also noted that DTE has no significant effect on earnings management. Interestingly, Sari and Afandi [40] found that DTE negatively affects earnings management, while current tax expense has a positive effect. Likewise, Saodah and Saefurahman [23] identified a partially negative relationship between DTE and earnings management.

The relationship between TP and earnings management also yields inconsistent findings. Salah [11] indicated that TP affects earnings management indirectly through net deferred tax liabilities rather than directly; Bunaca [37] and Handayani et al. [24] found no significant effect of TP on earnings management. In contrast, Permatasari and Trisnawati [6] argue that TP plays a regulatory role in earnings management. These findings suggest that research on the mediating and regulatory effects of deferred taxes is limited but increasingly attracting attention.

Studies emphasizing the institutional context show that results vary across countries and sectors. Augustine et al. [25] found that deferred taxes are positively associated with profitability in Nigeria yet have no significant effect on cash flow. Mura [26] highlighted that private firms in Italy opportunistically use deferred taxes to gain advantages in debt contracts. Moniz et al. [41] showed that highly leveraged firms recognize more DTA and the 2008 global financial crisis did not alter these behaviors. These findings indicate that the role of deferred taxes in earnings management is sensitive to institutional, sectoral, and macroeconomic conditions.

This comprehensive literature review indicates that the relationship between DTA, DTE and TP with earnings management is empirically highly debated. The differences in findings are attributed to studies being conducted in different countries, periods, and institutional settings [41]. Furthermore, many studies have examined these three variables—DTA, DTE and TP—separately rather than jointly. Importantly, the majority of existing studies rely on static econometric methods, which are insufficient to capture short-term dynamics and long-term equilibrium relationships between variables.

Consequently, there is a need for advanced econometric approaches capable of simultaneously testing the short- and long-term effects of deferred tax items on earnings management. In this context, the Panel ARDL model provides an appropriate methodological framework due to its applicability even when variables have different integration orders (I (0) and I (1)) and its capacity to examine both short- and long-term relationships within a single model.

This study aims to fill the existing gap in literature by analyzing the effects of DTA, DTE and TP on earnings management from a dynamic and long-term perspective, thereby contributing to academic literature.

## 3. Materials and methods

The present study used information from the financial statements of 20 companies operating in the food, beverage, and tobacco sectors listed on the Borsa Istanbul (BIST) in Türkiye and traded in the BIST 30 Index in the years 2013–2022. The data were obtained from Türkiye's Public Disclosure Platform (PDP), which publishes the notifications required to be disclosed to the public under the Türkiye Capital Market and Stock Exchange Law. Earnings management was used as the dependent variable, and tax planning, DTE, and DTA were used as independent variables. Firm Size and Leverage are included as control variables. Table 1 lists the variables used in the study, their symbols, and the source from which they were acquired.

Accrual earnings management, as measured by the modified Jones model to identify earnings management, is the dependent variable in this study. Managers use accrual earnings management to influence the output of the accounting system and to change the timing or format structure of business, investment, or financial activities. The calculation of earnings management is based on a modified version of the Jones model proposed by Dechow et al. [32]. For discretionary accruals, this model is estimated with at least 15 observations for each year and for each sector-year grouping. This requirement was fulfilled by the 20 samples chosen in the study. Discretionary accruals are determined by calculating non-discretionary accruals as a proportion of total accruals in the Modified Jones Model. The model estimate is shown in Equation 1 [11]:

$$TACCt = \Delta CAt - \Delta CASH - \Delta CLt + \Delta DCLt - DEPt \tag{1}$$

TACCt is the total accruals for the period t;
$\Delta CAt$ is the change in current assets for the period t;

**Table 1. Variables and descriptions.**

| Variable | Symbol | Resource |
|---|---|---|
| Earnings Management | TACCF | PDP * |
| Tax Planning | TP | PDP * |
| Deferred Tax Assets | DTA | PDP * |
| Deferred Tax Expenses | DTE | PDP * |
| Total Accruals | TACC | PDP * |
| Current Assets | CA | PDP* |
| Cash and Cash Equivalents | CASH | PDP* |
| Current Liabilities | CL | PDP* |
| Short Term Debt İncluded in Current Liabilities | DCL | PDP* |
| Depreciation And Amortization | DEP | PDP* |
| Revenues | REV | PDP* |
| Receivables | REC | PDP* |
| Gross Property Plant And Equipment | PPE | PDP* |
| Total Assets | TA | PDP* |
| Net Income | NI | PDP* |
| Net Income Before Tax | NIT | PDP* |
| Firm Size | SIZE | PDP* |
| Leverage | LEV | PDP* |

* Calculated by the author using Public Disclosure Platform data (Public Disclosure Platform, https://www.kap.org.tr/tr/, accessed: 15.10.2024).

ΔCASH is the change in cash and cash equivalents for the period;

ΔCLt is the change in current liabilities for the period t;

ΔDCLt is the change in short term debt for the period t;

DEPt is the depreciation expense for the period t.

Total accruals are then calculated using the modified Jones model, as shown in Equation 2:

$$\frac{TACC_t}{A_{t-1}} = \alpha_1 \frac{1}{A_{t-1}} + \alpha_2 \frac{(\Delta REV_t - \Delta REC_t)}{A_{t-1}} + \alpha_3 \frac{PPE_t}{A_{t-1}} \varepsilon_t \tag{2}$$

$TACC_t$ is total accruals for the period t divided by total assets for the period t-1,

$\Delta REV_t$ is revenues for the period t less revenues for the period t-1;

$\Delta REC_t$ is receivables for the period t less receivables for the period t-1;

$PPE_t$ is tangible fixed assets for the period t;

$A_{t-1}$ is total assets for the period t-1;

$\alpha_1$, $\alpha_2$, and $\alpha_3$ are parameters to be estimated, being coefficients estimated via an ordinary least squares regression.

Deferred Tax Assets (DTA) are one of the independent variables. According to the TAS 12 Income Taxes Standard, the concept of deferred tax arises in the case of temporary differences between commercial profit and financial profit. Temporary differences are classified into deductible temporary differences and taxable temporary differences. In the case of deductible temporary differences, DTA arise [14,19] the amount of DTA are recorded when it is probable that future tax benefits will be realized. Therefore, DTA should be considered when estimating the probability of their realization. DTA represent the amount of tax that is recoverable in future periods due to deductible temporary differences, unused tax losses and carry forward of unutilized tax credits and exemptions [26]. In this study, DTA are calculated using Equation 3 [2]:

$$DTA_t = \frac{\Delta DTA_t}{DTA_{t-1}} \tag{3}$$

$DTA_t$ is deferred tax assets of firm for the period t;

$\Delta DTA_t$ is change in deferred tax assets of firm for the period t;

$DTA_{t-1}$ is deferred tax assets of firm for the period t-1;

The second independent variable in the study is Deferred Tax Expense (DTE). Deferred tax expense arises from temporary differences between pre-tax and taxable income [2]. In this research, deferred tax expense is calculated as in Equation 4:

$$DTE_t = \frac{\Delta DTE_t}{TA_{t-1}} \tag{4}$$

$DTE_t$ is the amount of deferred tax expense of firm for the period t;

$\Delta DTE_t$ is change in DTE of firm for the period t;

$TA_{t-1}$ is total assets of firm t for the period t-1.

Tax planning (TP) is an important process that enables firms to minimize their tax burden within legal boundaries and thereby increase their after-tax profitability, and it is considered one of the independent variables in this study. TP is defined as the process of organizing taxes in such a way that companies can take advantage of various loopholes in tax regulations to pay the minimum amount of tax [24]. In this context, the tax retention rate (TRR) is used to evaluate the effectiveness of tax management in firms' financial reporting. TRR is stated to provide an appropriate indicator for

measuring TP effectiveness is calculated by dividing the net income of year $t$ by the pre-tax net income of year $t$–1 [11]. This ratio indicates the firm's capacity to generate after-tax income in the current period compared to its pre-tax performance in the previous period and is used as an indirect measure of TP effectiveness. Therefore, when the ratio is high, it can be inferred that the firm has been able to reduce its tax burden and maximize its net profit through its TP strategies [42,43]. Accordingly, the equation is given in (5):

$$TP = \frac{Net\ Income_t}{Net\ İncome\ Before\ Tax_{t-1}}$$

(5)

In this study, the earnings management variable TACCF was obtained by estimating the modified Jones model given in Equation 2. The relationship between earnings management and the DTE, DTA, and TP variables is analyzed using Model 1 (Equation 6) and Model 2 (Equation 7), as shown below.

$$Model\ 1: TACCFit = a1 + a2DTEit + a3DTAit + uit$$

(6)

$$Model\ 2: TACCFit = b1 + b2DTEit + b3DTAit + b4TPit + uit$$

(7)

## 4. Empirical analysis and results

Since the present study used data from 20 companies listed in the Borsa Istanbul BIST 30 index for the years 2013–2022, the analyses were conducted using panel data analysis methods. In this context, the presence of cross-sectional dependence in the variables was examined, and the stationarity of the series was then tested using appropriate unit root tests. Whether the coefficients obtained in the models are homogeneous for all cross-sections was checked using the slope homogeneity test, and the models were then estimated using the appropriate coefficient estimation method. This section presents theoretical information on the cross-sectional dependence test, the homogeneity test, and the panel CIPS unit root test and explains the results.

For cross-sectional dependence tests, the preferred tests differ according to the size of the time and cross-sectional dimensions. If the time dimension is larger than the cross-sectional dimension, the Breusch and Pagan [44] LM test is applied. However, if the cross-sectional dimension is larger than the time dimension, the scaled LM test of Pesaran [45] is applied. If the time and cross-sectional dimensions are infinite, Pesaran's [45] scaled LM test is applied. However, in this test, when the cross-sectional dimension is large and the time dimension is small, there are deviations due to size bias [46]. Therefore, Pesaran [45] recommends using the CD test when the cross-sectional dimension is large and the time dimension is small. Since the cross-sectional dimension is larger than the time dimension in this study, the CD test in Equation 8, proposed by Pesaran [45], was used.

$$CD_{PES} = \sqrt{\frac{2T}{n(n-1)}} \left( \sum_{i=1}^{n-1} \sum_{j=i+1}^{n} \hat{\rho}_{ij} \right)$$

(8)

Table 2 shows the results of the cross-sectional dependence tests of the variables:

Since the cross-sectional dimension in this study is larger than the time dimension, the CD test results of Pesaran [45] are used in Equation 8. According to the CDPES test results, there is cross-sectional dependence in all variables except DTE, TP and LEV. In this case, first-generation unit root tests were used to examine the stationarity of the DTE, TP and LEV variables, while second-generation unit root tests that consider cross-sectional dependence were used for

**Table 2. Cross-section dependence test results.**

| Variables | $LM_{BP}$ | $LM_{PES}$ | $LM_{BCadj}$ | $CD_{PES}$ |
|---|---|---|---|---|
| TA | 1097.635 [0.000] | 46.560 [0.000] | 45.449 [0.000] | 30.851 [0.000] |
| PPE | 316.053 [0.000] | 6.466 [0.000] | 5.355 [0.000] | 3.347 [0.000] |
| ΔR(REV-REC) | 873.523 [0.000] | 35.064 [0.000] | 33.952 [0.000] | 20.116 [0.000] |
| DTE | 280.334 [0.000] | 4.634 [0.000] | 3. 522 [0.000] | −0.930 [0.352] |
| DTA | 272.430 [0.000] | 4.228 [0.000] | 3. 117 [0.000] | 1.669 [0.094] |
| TP | 226.021 [0.037] | 1.847 [0.064] | 0. 736 [0.461] | 0.245 [0.805] |
| TACC | 300.444 [0.000] | 5.665 [0.000] | 4.554 [0.000] | 8.152 [0.000] |
| TACCF | 1134.322 [0.000] | 48.442 [0.000] | 47.331 [0.000] | 32.185 [0.000] |
| SIZE | 1641.608 [0.000] | 74.465 [0.000] | 73.354 [0.000] | 40.442 [0.000] |
| LEV | 398.058 [0.000] | 10.673 [0.000] | 9.562 [0.000] | −0.243 [0.807] |

other variables. Nevertheless, the manifestation of cross-sectional dependence in either weak or strong forms plays a critical role in determining the appropriate coefficient estimation method in econometric modeling. Weak cross-sectional dependence is defined as the case where, despite an increasing number of cross-sectional units, the aggregate effect of common factors on the dependent variable remains constant. Conversely, when the impact of common factors on the dependent variable intensifies in line with the growing number of cross-sectional units, this is referred to as strong cross-sectional dependence. Neglecting the existence of strong cross-sectional dependence in either the dependent or independent variables may lead to biased estimators and inconsistent results. By contrast, in the presence of weak cross-sectional dependence, it is not strictly necessary to employ specialized estimation techniques that explicitly account for such dependence [47]. Accordingly, the strength of cross-sectional dependence was tested for the variables [48]. The test statistics and probability values are reported as follows: TACCF (stat.: −0.44, prob.: 0.661), DTE (stat.: −0.38, prob.: 0.704), DTA (stat.: −1.64, prob.: 0.100), TP (stat.: 1.01, prob.: 0.313), SIZE (stat.: −0.76, prob.: 0.446), and LEV (stat.: −0.69, prob.: 0.490). These results collectively confirm the validity of the null hypothesis of 'weak cross-sectional dependence' across all variables.

For the panel unit root test of Levin et al. [49], which can be applied in the presence of cross-section independence and homogeneity across units in the panel, the t-statistics are obtained by applying the general procedure in Equation 9.

$$\Delta Y_{it} = \delta Y_{it-1} + \sum L = 1 Pi\theta_{iL}\Delta Y_{it-L} + \alpha_{mi}d_{it} + \varepsilon_{it}, \ m = 1, 2, 3. \tag{9}$$

Since Pi is not known here, a three-step procedure is used to apply the test by calculating the pooled t-statistics in Equation 10. The test statistics are compared with the results of the table values in Levin et al. [49], and if the null hypothesis is not accepted, it is concluded that each series in the panel does not follow a unit root process.

$$t_{\delta}^* = t_{\delta} - N\widetilde{T}\hat{S}_N\hat{\sigma}_{\widetilde{\varepsilon}}^{-2}STD(\hat{\delta})\mu_{m\widetilde{T}}^*\sigma_{m\widetilde{T}}^* \tag{10}$$

Here, $\mu^*_{m\widetilde{T}}$ ve $\sigma^*_{m\widetilde{T}}$ are the mean and standard deviation correction values.

When there is cross-sectional dependence among the series, it is appropriate to use a test that accounts for this situation. Equation 12 shows the Panel CIPS test proposed by Pesaran [50] to be used in the presence of cross-sectional dependence in the series. In this test, the CADF regression is estimated using the extended version of the ADF regression with lagged cross-section averages as shown in Equation 11, and the CIPS statistic is then obtained by averaging the estimated CADF values.

$$\Delta Y_{it} = a_i + b_i Y_{i,t-1} + c_i \overline{Y}_{t-1} + d_i \Delta \overline{Y}_t +$$

(11)

$$CIPS(N, T) = N^{-1} \sum_{i=1}^{N} CADF_i$$

(12)

Table 3 presents the unit root test results.

Analyzing the unit root test results presented in Table 3, the variables PPE, ΔR(REV-REC), DTA, TACC and TP are stationary at level in both constant and constant and trend models. The variables TA, TACCF, DTE, SIZE and LEV are stationary at first difference.

To examine whether a change in one of the units considered in the panel data analysis affects other units at the same level, the homogeneity of the slope coefficients should be examined. This observation also helps determine the appropriate estimation method. In this context, the slope homogeneity test proposed by Pesaran and Yamagata [51] is used.

**Table 3. Results of unit root test.**

**CIPS Unit Root Test**

| Variable | Constant | | Constant and Trend | |
|---|---|---|---|---|
| | t st. | Prob. | t st. | Prob. |
| TA | −2.237* | <0.10 | −2.777 | >=0.10 |
| PPE | −2.575** | <0.05 | −4.248*** | <0.01 |
| ΔR(REV-REC) | −2.561** | <0.05 | −5.685*** | <0.01 |
| DTA | −3.355*** | <0.01 | −5.498*** | <0.01 |
| TACC | −2.511** | <0.05 | −19.214*** | <0.01 |
| TACCF | −1.695 | >=0.10 | −2.360 | >=0.10 |
| SIZE | −1.849 | >=0.10 | −2.461 | >=0.10 |
| dTA | −4.895*** | <0.01 | −5.535*** | <0.01 |
| dTACCF | −4.496*** | <0.01 | −4.675*** | <0.01 |
| dSIZE | −2.762*** | <0.01 | −4.374*** | <0.01 |
| **LLC Unit Root Test** | | | | |
| DTE | −0.010 | >=0.10 | −2.173** | <0.05 |
| TP | −17.492*** | <0.01 | −43.833*** | <0.01 |
| LEV | −0.540 | >=0.10 | −2.752*** | <0.01 |
| dDTE | −5.510*** | <0.01 | −11.095*** | <0.01 |
| dLEV | −1.709** | <0.05 | −10.351*** | <0.01 |

* 10%, ** 5%, and *** 1% indicate the significance level of the coefficients. Critical values for the CIPS test were determined as −2.64 (1%), −2.33 (5%), and −2.18 (10%) in the constant model of the test. Critical values were determined as −3.46 (1%), −3.02 (5%), and −2.82 (10%) in the constant and trend model of the test. Pesaran's [50] article was used to calculate critical values.

According to the test's null hypothesis, the slope coefficients are homogeneous. The alternative hypothesis is that the slope coefficients are heterogeneous. The test is presented in equations 13 and 14 [51].

$$\widetilde{\Delta} = \sqrt{N}\left(\frac{N^{-1}\widetilde{S} - k}{\sqrt{2k}}\right) \tag{13}$$

$$\widetilde{\Delta}_{adj} = \sqrt{N}\left(\frac{N^{-1}\widetilde{S} - E(\hat{Z}_{iT})}{\sqrt{Var(\hat{Z}_{iT})}}\right) \tag{14}$$

N is the number of cross-sections, S is Swamy's test statistic, and k is the number of regressors.

The test results are presented in Table 4:

According to the results presented in Table 4, the slope coefficients for Model 1 and Model 2 are homogeneous. This situation indicates that the regression coefficients to be calculated for Model 1 and Model 2 variables do not differ from unit to unit.

If the variables are not stationary at the level according to the unit root test results, cointegration analysis can be used to examine the relationship between them. In this context, the pooled mean group (PMG) estimation method proposed by Pesaran et al. [52] for estimating the panel autoregressive distributed lag (ARDL) model was used in this study. The basic assumption of this model for estimating the coefficients is that the short-run coefficients are heterogeneous, but the long-run coefficients are homogeneous (i.e., the same for all units in the panel) [53]. This test is suggested when the independent variables have different levels of integration, but the dependent variable is I (1) and does not consider the cross-sectional dependence between series [54].

The panel ARDL model showing long-run relationships is shown in Equation 15:

$$Y_{it} = \alpha_i + \sum_{j=1}^{p} \beta_{ij} Y_{it-j} + \sum_{j=0}^{q} \gamma_{ij} X_{it-j} + \sum_{j=0}^{k} \delta_{ij} A_{it-j} + \sum_{j=0}^{m} \theta_{ij} B_{it-j} + \varepsilon_{it} \tag{15}$$

i=1,..., N is the number of cross-sections, t=1,..., T is the time dimension, and ε is the error term. The optimal lag lengths for each variable and unit are denoted by p, q, k, and m [55]. Equation 15 is then written in the form of an error correction model to estimate the short- and long-run parameters. The error correction forms of panel ARDL models are defined as in Equation 16:

$$\Delta Y_{it} = \alpha_i + \lambda_i Y_{it-1} + \gamma'_i X_{it} + \delta'_i A_{it} + \theta'_i B_{it} + \sum_{j=1}^{p-1} \beta''_{ij} \Delta Y_{it-j} + \sum_{j=0}^{q-1} \gamma''_{ij} \Delta X_{it-j} + \sum_{j=0}^{k-1} \delta''_{ij} \Delta A_{it-j} + \sum_{j=0}^{m-1} \theta''_{ij} \Delta B_{it-j} + \varepsilon_{it} \tag{16}$$

In Equation 16, the term λi represents the expected negative error correction coefficient and is shown as $\lambda_i = -\left(1 - \sum_{j=1}^{p} \beta_{ij}\right)$, $\gamma'_i = \sum_{j=0}^{q} \gamma_{ij}$, $\delta'_i = \sum_{j=0}^{k} \delta_{ij}$, $\theta'_i = \sum_{j=0}^{m} \theta_{ij}$, Here Δ denotes the first order difference operator; $\gamma'$, $\delta'$, $\theta'$ are the long-run coefficients; and $\beta''$, $\gamma''$, $\delta''$, $\theta''$ are the short-run coefficients [56].

Table 4. Homogeneity test results.

|  |  | Stat. | Prob. |
|---|---|---|---|
| Model 1 | Delta_tilde | 0.933 | 0.351 |
|  | Delta_tilde_adj | 1.204 | 0.228 |
| Model 2 | Delta_tilde | 0.221 | 0.825 |
|  | Delta_tilde_adj | 0.312 | 0.755 |

The results are shown in Tables 5 and 6 using the PMG estimator proposed by Pesaran et al. [52] to estimate Equation 16:

According to the results presented in Table 5, the value of the error correction parameter (ECM) (−0.381) is negative and statistically significant at the 1% level, indicating the existence of a cointegration relationship between the variables. In the long run, the DTE and DTA variables have a negative and significant impact on earnings management. The value of the error correction parameter (−0.381) indicates the speed with which the short-term deviations of the series stabilize in the next period. Approximately 38% of the imbalances that occur in one period are corrected in the next period, and the long-run equilibrium is reached after about three periods. The fact that DTE are meaningful and positive in the short run suggests that it positively affects profit management in the short run. However, as it increases tax expenditure in subsequent periods, it has the opposite effect on earnings management.

According to the results presented in Table 6, the value of the error correction parameter (ECM) (−0.419) is negative and statistically significant at the 1% level, showing the existence of a cointegration relationship between the variables. In the long run, the variables DTE, DTA and TP have a negative and significant effect on earnings management. The value of the error-correction parameter (−0.419) indicates that about 42% of the imbalances that occur in one period are corrected in the next period, reaching the long-run equilibrium after about two and a half periods. In all models, the lag lengths were determined through the automatic selection procedure in the EViews software. To ensure robustness, the Akaike Information Criterion (AIC) value for Model 1 was calculated as 9.784, and the optimal lag length was identified as one. The robustness of the estimation results was further examined by re-estimating the models: firm size and leverage ratio were sequentially incorporated as control variables into Model 1 and Model 2, respectively. In addition, a subsample of 15 firms was constructed by shortening the time period and the analysis was replicated. The robustness check results are reported in Table 7.

The robustness analysis conducted by incorporating control variables and utilizing a subsample yields results consistent with those of the baseline Model 1 and Model 2. For the control-variable-augmented Model 1, all coefficient signs remain unchanged. When firm size is included as a control variable, DTA becomes insignificant, whereas when leverage

**Table 5. PMG estimation results for Model 1.**

| Model 1 ARDL (1,1,1) | | Coeff. | Test Stat. | Prob. |
|---|---|---|---|---|
| **Long Run Estimation** | DTE | −13.261 | −4.581 | 0.000 |
| | DTA | −0.0002 | −2.450 | 0.015 |
| **Short Run Estimation** | ECM | −0.381 | −3.277 | 0.001 |
| | ΔDTE | 4.386 | 2.590 | 0.010 |
| | ΔDTA | −0.016 | −0.803 | 0.423 |
| | C | 0.119 | 3.735 | 0.000 |

**Table 6. PMG estimation results for Model 2.**

| Model 2 ARDL (1,1,1,1) | | Coeff. | Test Stat. | Prob. |
|---|---|---|---|---|
| **Long Run Estimation** | DTE | −4.089 | −3.522 | 0.000 |
| | DTA | −0.0003 | −4.018 | 0.000 |
| | TP | −0.025 | −2.303 | 0.023 |
| **Short Run Estimation** | ECM | −0.419 | −3.985 | 0.000 |
| | ΔDTE | 0.157 | 0.029 | 0.976 |
| | ΔDTA | 0.004 | 0.542 | 0.588 |
| | ΔTP | 0.064 | 0.304 | 0.761 |
| | C | 0.116 | 5.454 | 0.000 |

**Table 7. Long run PMG estimation results for Model 1 and Model 2 with control variables.**

| | Model 1 + Control | | | | Model 2 + Control | | | | |
|---|---|---|---|---|---|---|---|---|---|
| | DTE | DTA | SIZE | LEV | DTE | DTA | TP | SIZE | LEV |
| **Coeff.** | −4.610 | −1.83E-05 | 0.060 | | −9.656 | −9.46E-05 | 0.0003 | 0.017 | |
| | −0.566 | −0.00056 | | −0.282 | −1.308 | −0.0008 | −0.013 | | −0.175 |
| **Test Stat.** | −5.132 | −0.197 | 4.410 | | −24.137 | −1.820 | 0.424 | 2.295 | |
| | −0.414 | −3.387 | | −2.671 | −714.19 | −4945.97 | −183.14 | | −1060.1 |
| **Prob.** | 0.000 | 0.844 | 0.000 | | 0.000 | 0.075 | 0.673 | 0.026 | |
| | 0.680 | 0.001 | | 0.009 | 0.000 | 0.000 | 0.000 | | 0.000 |

is included, DTE is rendered insignificant. In both specifications of Model 1, however, the control variables themselves are statistically significant. In the control-variable-augmented Model 2, the coefficient signs of all variables remain the same, with the exception of TP. Specifically, TP becomes insignificant when firm size is included as a control variable, while in the model incorporating leverage, all variables are statistically significant. Hence, both control variables are found to be significant in Model 2. Furthermore, the magnitude of coefficients in the control-variable-augmented models closely resembles those obtained in the baseline models. In Model 1 with firm size as a control variable, the error correction term (ECM) is estimated at −0.874, which is statistically significant at the 1% level. In Model 1 with leverage as a control variable, the ECM is estimated at −0.601, also statistically significant. Similarly, in Model 2 with control variables, the ECM coefficients are negative and statistically significant. Overall, these findings confirm the existence of a long-run cointegration relationship among the variables even when control variables are introduced.

## 5. Discussion and conclusion

Due to differences between accounting standards and tax laws, there are two earnings figures. The literature demonstrates that managers can engage in earnings management practices by exploiting flexibilities in accounting standards. The treatment of deferred taxes in the IAS 12 Income Taxes is one of the relevant flexibilities. This study sought to investigate the extent to which DTA, DTE and TP affect earnings management in Türkiye.

The study used data from the financial statements of 20 companies in the Borsa Istanbul BIST 30 index operating in the food, beverage, and tobacco sectors for the years 2013–2022. The analyses were conducted using panel data analysis methods. First, the presence of cross-sectional dependence in the variables was examined, and the stationarity of the series was then tested using appropriate unit root tests. Whether the coefficients obtained in the models were homogeneous for all cross-sections was checked using the slope homogeneity test, and the models were then estimated using the appropriate coefficient estimation method. To assess the robustness of the estimation results, a robustness check was performed by incorporating control variables into the established models and employing a subsample comprising 15 firms.

According to the results obtained in the model constructed with DTE and DTA, the value of the error correction parameter (ECM) (−0.381) is negative and statistically significant, indicating the existence of a cointegration relationship between the variables. In the long run, the variables DTE and DTA have a negative and significant effect on earnings management. The value of the error correction parameter (−0.381) indicates the speed with which the short-term deviations of the series stabilize in the next period. Approximately 38% of the imbalances that occur in one period are corrected in the next period, and the long-run equilibrium is reached after about three periods. The fact that DTE are significant and positive in the short run suggests it positively affects profit management in the short run. However, as DTE increase tax expenditure in subsequent periods, earnings management will be affected in the opposite direction.

This study is in line with previous research by Sari [40], which states that deferred tax expense has a negative impact on earnings management. This implies that the lower the profit, the greater the opportunity to engage in earnings

management. DTE can affect earnings management, indicating that companies may tend to engage in earnings management to avoid losses or profit declines.

With the addition of TP to the model as another variable that may affect earnings management, this study reveals a cointegration relationship between the variables. In the long run, the variables DTE, DTA and TP have a negative and significant effect on earnings management. The value of the error correction parameter (−0.419) indicates that about 42% of the imbalances that occur in one period are corrected in the next period, reaching the long-run equilibrium after about two and a half periods.

As the literature and the results of this study show, DTA, DTL and TP are related to earnings management. Companies manage their earnings via deferred taxes, tax planning, and other accounting flexibilities. To ensure that users of financial statements have access to accurate and reliable information, measures should be taken to reduce earnings management practices. To this end, national accounting standards should be improved and updated in line with International Financial Reporting Standards (IFRS). In particular, the concepts and language of accounting standards should be standardized, alternative accounting policies and methods should be reduced, and management's accounting estimates and judgments should be minimized. Efforts should also be made to harmonize tax legislation and accounting standards.

## Supporting information

**S1 Data. Taxaccounting data.**
(XLSX)

## Author contributions

**Conceptualization:** Meral Gündüz.

**Data curation:** Meral Gündüz.

**Formal analysis:** Meral Gündüz.

**Investigation:** Meral Gündüz.

**Methodology:** Meral Gündüz.

**Resources:** Meral Gündüz.

**Software:** Meral Gündüz.

**Writing – original draft:** Meral Gündüz.

**Writing – review & editing:** Meral Gündüz.

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
