## [Decision Letter · Decision Letter 0]

22 Sep 2025

Dear Dr. Meral Gündüz,

Thank you for submitting your manuscript to PLOS ONE. After careful consideration, we feel that it has merit but does not fully meet PLOS ONE’s publication criteria as it currently stands. Therefore, we invite you to submit a revised version of the manuscript that addresses the points raised during the review process.

Following a thorough review by two reviewers and an academic editor, it must be noted that the manuscript requires significant revision. In particular, further clarifications are required and definitions need to be improved. The literature review should be expanded. It is essential to clarify the study's contribution to existing research in the field. Please ensure that you implement all of the reviewers' comments and resubmit the revised manuscript.

Reviewer #1:

Thank you for the opportunity to read and review your paper. I found it very interesting by the subject that you have chosen, that is suitable and very interesting in the current knowledge of accounting research. Also, I appreciate the scientific approach that you have chosen, but I would like to better present your research hypotheses and research objectives. The references are properly selected, the statistical results are correctly interpreted with high level of phenomenon understanding, and conclusions are suitable. In my opinion this paper could contribute with a high level of scientificness to the current level of knowledge in the field and I encourage the author to continue the research in the field.

Reviewer #2:

This manuscript under review focuses on firms included in the BIST 30 index of the Istanbul Stock Exchange (Turkey) and uses financial data from 2013 to 2022. The authors employ a panel ARDL model to examine the relationships between deferred tax assets (DTA), deferred tax expenses (DTE), tax planning (TP), and earnings management. The study attempts to combine heteroskedasticity and cross-sectional dependence tests in panel data with PMG estimation. The overall logical structure is basically coherent; however, several issues need to be addressed. Detailed comments are as follows:

Comment 1. In the manuscript, TP is proxied by “Net Incomeₜ / Net Income Before Taxₜ₋₁” (Equation 5), but the theoretical rationale linking this indicator to tax planning is not clearly explained.

Comment 2. Earnings management is influenced by various factors such as firm size, leverage, and growth opportunities. However, Models 1 and 2 include only the main explanatory variables (DTA, DTE, TP) without controlling for these potential confounders. The authors should justify the choice of control variables and perform robustness checks including them to ensure the reliability of the results.

Comment 3. Model 1 is specified as ARDL (1,1,1) and Model 2 as ARDL (1,1,1,1), but the manuscript does not explain how the lag lengths were determined.

Comment 4. Table 2 indicates that DTE and TP do not exhibit cross-sectional dependence, whereas other variables do. The manuscript only differentiates between “first-generation / second-generation unit root tests” but does not clarify how remaining cross-sectional dependence is addressed in the PMG estimation.

Comment 5. In the keywords, the term “Deferred tax expensive” appears incorrect. It should be revised to “Deferred tax expense” to accurately reflect the concept.

Comment 6. The reference formatting should be standardized according to the journal’s style.

Comment 7. It is recommended to include the following references to improve literature coverage and theoretical grounding.

(i)Zhong, Q., Song, Q. & Lee, CC. Managing crash risks through supply chain transparency: evidence from China. Financ Innov 10, 126 (2024). https://doi.org/10.1186/s40854-024-00633-3

(ii)Sun, Y., Liu, L., Xu, Y. et al. Alternative data in finance and business: emerging applications and theory analysis (review). Financ Innov 10, 127 (2024). https://doi.org/10.1186/s40854-024-00652-0

(iii)Kweh, Q.L., Lu, WM., Tone, K. et al. Evaluating the resource management and profitability efficiencies of US commercial banks from a dynamic network perspective. Financ Innov 10, 19 (2024). https://doi.org/10.1186/s40854-023-00531-0

We look forward to receiving your revised manuscript.

Kind regards,

Thomas Kollruss, Prof. Dr.

Academic Editor

PLOS ONE

Additional Editor Comments:

Reviewers' comments:

Reviewer's Responses to Questions

**Comments to the Author**

1. Is the manuscript technically sound, and do the data support the conclusions?

Reviewer #1: Yes

Reviewer #2: Yes

2. Has the statistical analysis been performed appropriately and rigorously?

Reviewer #1: Yes

Reviewer #2: Yes

3. Have the authors made all data underlying the findings in their manuscript fully available?

Reviewer #1: Yes

Reviewer #2: Yes

4. Is the manuscript presented in an intelligible fashion and written in standard English?

Reviewer #1: Yes

Reviewer #2: Yes

Reviewer #1: Dear Author,

Thank you for the opportunity to read and review your paper. I found it very interesting by the subject that you have chosen, that is suitable and very interesting in the current knowledge of accounting research. Also, I appreciate the scientific approach that you have chosen, but I would like to better present your research hypotheses and research objectives. The references are properly selected, the statistical results are correctly interpreted with high level of phenomenon understanding, and conclusions are suitable. In my opinion this paper could contribute with a high level of scientificness to the current level of knowledge in the field and I encourage the author to continue the research in the field.

Reviewer #2: PONE-D-25-12793

This manuscript under review focuses on firms included in the BIST 30 index of the Istanbul Stock Exchange (Turkey) and uses financial data from 2013 to 2022. The authors employ a panel ARDL model to examine the relationships between deferred tax assets (DTA), deferred tax expenses (DTE), tax planning (TP), and earnings management. The study attempts to combine heteroskedasticity and cross-sectional dependence tests in panel data with PMG estimation. The overall logical structure is basically coherent; however, several issues need to be addressed. Detailed comments are as follows:

Comment 1. In the manuscript, TP is proxied by “Net Incomeₜ / Net Income Before Taxₜ₋₁” (Equation 5), but the theoretical rationale linking this indicator to tax planning is not clearly explained.

Comment 2. Earnings management is influenced by various factors such as firm size, leverage, and growth opportunities. However, Models 1 and 2 include only the main explanatory variables (DTA, DTE, TP) without controlling for these potential confounders. The authors should justify the choice of control variables and perform robustness checks including them to ensure the reliability of the results.

Comment 3. Model 1 is specified as ARDL (1,1,1) and Model 2 as ARDL (1,1,1,1), but the manuscript does not explain how the lag lengths were determined.

Comment 4. Table 2 indicates that DTE and TP do not exhibit cross-sectional dependence, whereas other variables do. The manuscript only differentiates between “first-generation / second-generation unit root tests” but does not clarify how remaining cross-sectional dependence is addressed in the PMG estimation.

Comment 5. In the keywords, the term “Deferred tax expensive” appears incorrect. It should be revised to “Deferred tax expense” to accurately reflect the concept.

Comment 6. The reference formatting should be standardized according to the journal’s style.

Comment 7. It is recommended to include the following references to improve literature coverage and theoretical grounding.

(i)Zhong, Q., Song, Q. & Lee, CC. Managing crash risks through supply chain transparency: evidence from China. Financ Innov 10, 126 (2024). https://doi.org/10.1186/s40854-024-00633-3

(ii)Sun, Y., Liu, L., Xu, Y. et al. Alternative data in finance and business: emerging applications and theory analysis (review). Financ Innov 10, 127 (2024). https://doi.org/10.1186/s40854-024-00652-0

(iii)Kweh, Q.L., Lu, WM., Tone, K. et al. Evaluating the resource management and profitability efficiencies of US commercial banks from a dynamic network perspective. Financ Innov 10, 19 (2024). https://doi.org/10.1186/s40854-023-00531-0

**Do you want your identity to be public for this peer review?** For information about this choice, including consent withdrawal, please see our Privacy Policy

Reviewer #1: No

Reviewer #2: No

---

## [Author Response · Author response to Decision Letter 1]

29 Sep 2025

I am deeply grateful to the academic editor and reviewers for their insightful recommendations, which have helped refine this manuscript. I have carefully addressed all comments and implemented the suggested revisions to improve the quality of my manuscript. The revisions made taking into account the reviewers' comments are shown in red in the text and are listed below:

Rewiever 1:

I would like to extend my sincere gratitude to Reviewer 1 for their insightful comments and constructive feedback, as well as for their supportive and encouraging approach.

Rewiever 2:

Comment 1:

The theoretical rationale linking this indicator to tax planning is elaborated in the section pertaining to Equation 5.

Comment 2:

Robustness checks were conducted by incorporating firm size and leverage ratio as control variables, and the results are presented in the text.

Comment 3:

The procedure used to select the appropriate lag length are explained in the manuscript.

Comment 4:

The text explains how the presence and strength of cross-sectional dependence are accounted for in the PMG estimation framework.

Comment 5:

The keyword has been changed as suggested.

Comment 6:

The reference formatting has been revised to conform to the journal's style guidelines.

Comment 7:

The inclusion of new and recommended sources has served to strengthen the theoretical framework and literature review, thereby providing a more comprehensive academic foundation for the study.

---

## [Decision Letter · Decision Letter 1]

3 Nov 2025

The Impact of Tax Accounting and Planning on Earnings Management: Evidence from Panel ARDL Approach

PONE-D-25-12793R1

Dear Dr. Meral Gündüz, 

We’re pleased to inform you that your manuscript has been judged scientifically suitable for publication and will be formally accepted for publication once it meets all outstanding technical requirements.

Kind regards,

Prof. Dr. Thomas Kollruss

Academic Editor

PLOS ONE

Additional Editor Comments (optional):

All reviewer comments have been addressed. The article is ready for publication.

Reviewers' comments:

Reviewer's Responses to Questions

**Comments to the Author**

Reviewer #2: All comments have been addressed

Reviewer #3: All comments have been addressed

2. Is the manuscript technically sound, and do the data support the conclusions?

Reviewer #2: Yes

Reviewer #3: Yes

3. Has the statistical analysis been performed appropriately and rigorously?

Reviewer #2: Yes

Reviewer #3: Yes

4. Have the authors made all data underlying the findings in their manuscript fully available?

Reviewer #2: Yes

Reviewer #3: Yes

5. Is the manuscript presented in an intelligible fashion and written in standard English?

Reviewer #2: Yes

Reviewer #3: Yes

Reviewer #2: I have reviewed the revised manuscript submitted by the authors. I appreciate the significant efforts they have invested in this revision, which are clearly evident in the updated work. Having undergone a previous round of review, I find that the manuscript now demonstrates the necessary content depth, academic rigor, and novelty suitable for publication.

The authors have effectively addressed the key remarks and suggestions provided by the reviewers. These revisions have been appropriately and thoroughly integrated into the new version of the manuscript.

Key improvements observed:

Abstract: Has been effectively expanded to include the study's practical implications and limitations, resulting in a more comprehensive and informative summary.

Introduction: Now provides a clearer articulation of the specific research gap addressed, effectively distinguishing this work from previous, more general reviews.

Literature Review: Demonstrates not only broad thematic coverage but also a strengthened critical synthesis of past studies, explicitly acknowledging their theoretical and regional limitations.

Findings: Are well-organized and strongly supported by insightful sub-theme discussions.

Conclusion: Effectively reflects on the study's limitations and outlines specific, actionable directions for future research.

Overall, the revised manuscript exhibits clear academic rigor, relevance, and originality. I am satisfied with the authors' responses to the review comments and believe this revised version represents a significant improvement over the initial submission in terms of its completeness and robustness.

Therefore, I am pleased to recommend this manuscript for acceptance in its present form.

Reviewer #3: (No Response)

**Do you want your identity to be public for this peer review?** For information about this choice, including consent withdrawal, please see our Privacy Policy

Reviewer #2: No

Reviewer #3: No

---

## [Editor Report · Acceptance letter]

PONE-D-25-12793R1

PLOS ONE

Dear Dr. Gündüz,

I'm pleased to inform you that your manuscript has been deemed suitable for publication in PLOS ONE. Congratulations! Your manuscript is now being handed over to our production team.

Kind regards,

on behalf of

Professor Thomas Kollruss

Academic Editor

PLOS ONE